# Oxidative Stress and Performance after Training in Professional Soccer (European Football) Players

**DOI:** 10.3390/antiox12071470

**Published:** 2023-07-22

**Authors:** Michele Abate, Raffaello Pellegrino, Angelo Di Iorio, Vincenzo Salini

**Affiliations:** 1Vita-Salute San Raffaele University, 20132 Milan, Italy; 2Department of Scientific Research, Campus Ludes, Off-Campus Semmelweis University, Pazzallo, 6912 Lugano, Switzerland; 3Department of Innovative Technologies in Medicine & Dentistry, Gabriele D’Annunzio University, 66100 Chieti, Italy

**Keywords:** oxidative stress, soccer, muscle-performance, supplementation, mediators, free radicals, blood antioxidant potential

## Abstract

Vitamins, hormones, free radicals, and antioxidant substances significantly influence athletic performance. The aim of this study was to evaluate whether these biological mediators changed during the season and if this was associated with the rate of improvement in performance after training, assessed by means of a standardized test. Professional male soccer players took part in the study. Two evaluations were performed: the first in the pre-season period and the second at the mid-point of the official season, after about 6 months of intensive training and weekly matches. Blood levels of vitamins D, B12, and folic acid, testosterone and cortisol, free radicals, and antioxidant substances were measured. Two hours after breakfast, a Yo-Yo test was performed. The relationships between the biological mediators and the rate of improvement after training (i.e., the increase in meters run in the Yo-Yo test between the pre-season and mid-season periods) were evaluated by means of a linear mixed models analysis. Results: Eighty-two paired tests were performed. The athletes showed better performance after training, with an increase in the meters run of about 20%. No significant relationships between the vitamin and hormone values and the gain in the performance test were observed. Plasmatic levels of free radicals increased significantly, as did the blood antioxidant potential. An indirect relationship between oxidative stress and the improvement in performance was observed (free radicals β ± SE: = −0.33 ± 0.10; *p*-value = 0.001), with lower levels of oxidative stress being associated with higher levels of performance in the Yo-Yo test. Monitoring the measures of oxidative stress could be a useful additional tool for coaches in training and/or recovery programs tailored to each player.

## 1. Introduction

The response to training in athletes is influenced by several biological mediators (vitamins, hormones, oligoelements, etc.) [1]. Vitamin D (Vit D) is important in bone health, but recent research has also pointed out its essential role in extra-skeletal function, including muscle growth, immune and cardiopulmonary functions, and inflammatory modulation. Therefore, blood levels of Vit D influence athletic performance, and its supplementation appears to be an effective strategy to alleviate muscle damage and inflammation after exercise [2,3,4,5]. It is also claimed that iron, vitamin B12 and folic acid may have an ergogenic effect, and current research suggests that exercise may increase the requirements for these substances [6]. Therefore, athletes at risk of nutrient deficiency may show a decreased ability to perform exercise at high intensities [7,8]. Also, hormones may be important, and a decrease in the testosterone–cortisol ratio (expression of the prevalence of catabolism and anabolism) has been considered a harbinger of overtraining/overreaching [9,10,11]. More consistently, several studies have shown that after intense exercise, free radical (FR) production increases significantly [12]. Although skeletal muscles provide the major source of FR generation, due to their relatively large mass, it is feasible, however, that other tissues such as the lungs, heart, or white blood cells may contribute to the total body generation of FRs during physical activity [13]. Within physiological limits, FRs have positive effects, as they enhance muscle contraction, cell growth, and immune function; however, when produced in excess, they can interact with the cell membrane lipid bilayer, damaging a range of biological molecules, including not only lipids but also nucleic acids, carbohydrates, and proteins. Therefore, excessive FR production can have a negative impact on the immune system, the ability to perform and on health in general [14,15,16]. Increased FR production is, within certain limits, counteracted by a contemporary increase in antioxidant substances, which restore the redox state [17]. Both the extent of FR production as well the strength of athletes’ defenses against FRs varies greatly, depending both on exogenous and endogenous factors such as: the characteristics of physical exercise (duration, quality, and intensity), weather conditions, genetics, age, diet, and lifestyle habits (such as smoking and alcohol use) [18]. Elite athletes have a very strong antioxidant defense system; however, prolonged and strenuous exercise with an excessive increase in FR production can lead to oxidative damage, the development of fatigue, muscular injury, and overreaching/overtraining with the impairment of sport performance [19,20]. Several studies on this issue have been performed in soccer players, whose peculiar physical activities (sprints, high-intensity running, and sudden change of direction) involve both aerobic and anaerobic energy pathways, leading to oxidative stress (OS) [21,22].

Some studies have investigated the production of FRs and antioxidant substances after a single competitive event or a progressive maximal physical effort test; other studies have evaluated changes in basal redox status at different times in the competitive season or after a training program [11,23,24,25]. Overall, despite the differences related to the protocol design, these studies have shown that the redox status in professional soccer players is upregulated and may be altered according to the training period and the accumulation of training loads; moreover, during the winter, after several matches and repeated sessions of regular training, the blood levels of FRs as well as those of antioxidant potential appear to be significantly increased. Finally, to the best of our knowledge, only one study [26] has evaluated the relationship between OS and the level of performance during the competitive season, carrying out standardized athletic tests. The above-mentioned study showed relevant associations between the hormonal and redox parameters, match-accumulated time, and the performance in athletic tests.

Therefore, the aim of our research was to validate the results of this study; in addition, we aimed to evaluate whether the levels of some parameters of OS (FR production, blood antioxidant potential (BAP), and their ratio) and other biological mediators (hormones and vitamins) were associated with the rate of improvement in performance after training, assessed by means of a standardized performance test.

## 2. Materials and Methods

### 2.1. Subjects

This study was performed in accordance with the ethical standards as laid down in the 1964 Declaration of Helsinki and its later amendments. All participants were informed about the purpose and test procedures and gave written consent to participate in the study. Institutional ethics committee approval was given by the local IRB (CRRM; 2023; 06; 10; 01). The athletes enrolled were male football players (starting lines or substitutes), with at least five years’ experience in soccer at professional or semi-professional (juvenile) level, who trained regularly during the competitive season and who had not been injured and/or submitted to reduced individualized training programs in the last two months before the assessment. All of the athletes belonged to a team of the Second Italian Division (Serie B) and were studied during three agonistic seasons (2018–2021). Two evaluations were performed: the first at the beginning (August) and the other during the mid-season (February) period. During the competitive season, the athletes took part in five weekly training sessions (approximately 90 min each). The intensity, volume, and modalities of training were similar for all athletes. Moreover, they took part in one official game (occasionally two games) per week. A clinical control was regularly performed. All players were in good health and were not taking medication, nutritional supplements, or drugs that could influence the experimental protocol. Symptoms of possible overtraining/overreaching (fatigue, muscle pain, sleep disturbances, and depressed mood) were carefully and regularly checked. A qualified dietitian provided nutrition counselling to the players, their coaches, and trainers, guiding them in the correct interpretation of food labels and in the choice of well selected food. Their diet was made up of 55–65% carbohydrates, 12–15% protein, and less than 30% fat, with adequate intake of vitamins and minerals. In the off-season (holiday period), no dietary control was performed. 

### 2.2. Measures

Once the participants arrived at the training center (around 8–9 a.m.), the anthropometric measurements were performed. After shoes and heavy clothing were removed, weight (kg) and height (cm) were measured using an electronic scale and a stadiometer with an accuracy of ±0.1 kg and ±0.1 cm, respectively. Body mass index (BMI) was calculated for each subject by dividing their weight by the square of their height (kg/m^2^). Blood samples were collected from the antecubital vein in a seated position, after the athletes had been fasting overnight and at least 48 h after the last training session. Care was taken to test the players in the same temporal order to avoid any circadian variation in the measured variables. The blood samples were centrifuged for 10 min at a speed of 3500 rpm. The serum samples were then stored at −20 °C prior to analysis. FRs, BAP, testosterone, cortisol, vitamin B12, folic acid, hemoglobin, iron, ferritin, and creatin-phosphokinase were evaluated. For FRs, a free radical analytical system (Diacron, Grosseto, Italy) [27] was used. Briefly, in a pipette, 25 μL of the serum sample was mixed with acetic acid buffered solution (pH 4.8), and a chromogenic substrate was then added to the mixture. The mixture was centrifuged and then incubated in the thermostatic block of the system. The absorbance was recorded at 505 nm. The measurement of OS is expressed in U Carr (carratine units), where 1 U Carr corresponds to 0.08 mg/dL H_2_O_2_. The suggested FR upper limit is >300 U Carr. A value > 500 U Carr is considered to show the expression of severe OS [27]. The biological antioxidant activity of plasma was measured using a colored solution containing ferric Fe^3+^ ions bound to a chromogenic substrate, which is decolorized upon the reduction of Fe^3+^ to Fe^2+^ ions by the reducing power of the antioxidant activity of plasma added to the reaction solution. The intensity of decoloration can be measured photometrically at the wavelength of 505 nm. The normal value for BAP in healthy subjects is >2200 µM/L [27]. Free testosterone and cortisol were analyzed with assay kits from DRG diagnostics (DRG, International Inc., New York, NY, USA; Research Use Only, Testosterone CLA-4660, Cortisol CLA-4651, DRG, International Inc., New York, NY, USA). The intra- and inter-assay coefficients of variation (CVs) for testosterone were 3.7% and 5.6%, respectively. The intra- and inter-assay CVs for cortisol were 4.0% and 5.7%, respectively. All assays were performed in duplicate. The range of normal values is 2.6–8.7 ng/mL for testosterone and 6.4–21 µg/dL for cortisol. A testosterone–cortisol ratio < 0.75 is considered a risk factor of overtraining. The simultaneous quantitative determination of vitamin B12 and folate were measured by an RIA assay (Simul TRAC-SNB Radioassay Kit, ICN Diagnostics Division, Orangeburg NY, USA). Vit D levels were measured using an enzyme immunoassay (OCTEIA 25-Hydroxy Vitamin D kit, Immunodiagnostic Systems, Inc., Fountain Hills, AZ, USA). Vit D levels > 30 ng/dL are considered as sufficient, between 20–30 ng/mL is considered as insufficient, and between 10–20 ng/mL is considered as deficient. Normal values of vitamin B12 and folic acid are 160–950 picograms per ml (pg/mL) and 3–17 nanograms per milliliter (ng/mL), respectively.

### 2.3. Yo-Yo Test

After blood withdrawal the subjects had a personalized standard breakfast [28]. Approximately 2 hours later, after a 30-min warm-up, the athletes performed the Yo-Yo Intermittent Recovery Test Level 2 supervised by the same experienced investigator [29,30,31]. To minimize circadian rhythms and climate-related factors, the test was performed in similar environmental conditions (temperature: 18–20 °C; humidity: 50–60%) on an artificial turf, which is almost unaffected by weather conditions. The test requires repeated 2 × 20 m shuttle runs between a start and finish line, at a progressively increased speed controlled by an audio metronome from a calibrated CD player. There is a 10 s period of active recovery (decelerating and walking back to the starting line) between runs. When a subject fails twice to reach the finishing line in time, the distance covered at that point is recorded and considered the test result. The test begins at 13 km/h speed, with subsequent speed increments, and usually lasts 5–15 min. This high intensity exercise with a large anaerobic contribution alongside a significant aerobic component is considered a reliable and valid measurement of match-related fitness performance in professional soccer athletes [29]. 

### 2.4. Statistical Analysis

The baseline characteristics were compared with mid-season assessment using Quantile regression, which is especially useful with heterogeneous data (i.e., for those variables that show tails and whereby the central location of the conditional distributions varies differently from the covariates). To test whether BAP, FRs, and their ratio were associated with the rate of improvement after training (i.e., the increase in meters run in the Yo-Yo test between the pre-season and mid-season periods), we performed linear mixed models (LMM) analysis; each parameter was considered as the independent variable (fixed effects), in different models; moreover, age and cortisol serum levels were considered in the models as potential confounders. Second-order analyses were also performed to verify the multiplicative effect of the interaction for the time of the study, and BAP, FRs (U CARR), and their ratio; moreover, as random effects, intercept and slope, with unstructured covariance, were also considered in the models. Model A reported the unconditional means model, which evaluated just the random effect for the intercept without any predictors; Model B reported the unconditional growth model, which considered the effect of time; and Models C, D, and E also included a second-order analysis with an interaction term; all models were also adjusted for C serum level. Akaike’s information criterion (AIC) was used to examine improvements in model fit; for all studied models, smaller values represent better fitting models [32]. The following estimates were derived from the analysis: γ_00_ = intercept of the average trajectory; γ_01_ = intercept of the trajectory for Age; γ_02_ = intercept of the trajectory for BAP; γ_03_ = intercept of the trajectory for U CARR; γ_04_ = intercept of the trajectory for BAP/U CARR; γ_01*03_ = intercept of the trajectory for Age*U CARR; γ_01*04_ = intercept of the trajectory for Age*BAP/U CARR; γ_10_ = slope of the average trajectory; γ_12_ = slope of the average trajectory for time*BAP; γ_13_ = slope of the average trajectory for time*U CARR; γ_14_ = slope of the average trajectory for time*BAP/U CARR; δ^2^_e_ = within-person variance components; and δ_20_ = in initial status variance components [33]. SAS version 9.4 for Windows (SAS Institute, Inc., Cary, NC, USA) was used for all data processing and statistical analyses. We set the level of statistical significance at *p* < 0.05 (2-tailed).

## 3. Results

In total, 42 athletes were studied during three agonistic seasons (2018–2021): 14 performed the tests in all three seasons, 12 performed the tests only in two seasons, and 16 performed the tests only in one season. Overall, 82 paired tests were evaluated. In Table 1, anthropometric measures and changes in the Yo-Yo performance test, as well in the laboratory parameters, between the pre-season and mid-season period are reported. In the mid-season period, a small (but not significant) decrease in BMI was registered; as expected, the performance in the Yo-Yo test improved significantly. Besides that, no changes in the plasma values of hemoglobin, iron, vitamin B12, and folic acid were observed; a decrease (but not statistically significant) in Vit D and testosterone was observed. On the contrary, plasma levels of cortisol and FRs (expressed as U CARR) increased statistically significantly, as well BAP. The BAP/U CARR ratio was reduced in the mid-season period (but not significantly). 

As can be seen in Table 2 (Model A), across individuals and time, the Yo-Yo test mean (in meters) was 893.49 ± 13.91 (*p*-value < 0.001), the variance within athletes was 21495 ± 3336 (*p*-value < 0.001), and the variance between athletes was 5313 ± 3000 (*p*-value = 0.04); an estimated 20% of the total variation in the test was attributed to differences the between subjects. When the time effect was considered in the analysis (Table 2, Model B), it could be demonstrated that the athletes performed less well in the Yo-Yo test in pre-season period compared to the mid-season period (β ± SE: −184.34 ± 10.42; *p*-value < 0.001). 

In Figure 1, the two performance test assessments are depicted: as can be seen, the two lines are parallel, but in the mid-season period, the athletes performed at a higher level compared to the pre-season period. 

Adjusting Model B for age and C serum levels, these variables were significantly associated with performance in the Yo-Yo test; the time effect was practically unchanged compared to the previous model (β ± SE: −207.92 ± 12.96; *p*-value < 0.001); for cortisol-serum level, the estimated effect (β ± SE) was −0.29 ± 0.13, *p*-value = 0.03, and the age effect was the equivalent to a decrease in performance of −12.37 ± 2.85 m per year of age (*p*-value < 0.001). A multiplicative effect for the interaction was not evident. This is also shown in Figure 2, where the course of association between the Yo-Yo test and age in the two periods of the study was plotted. It is evident that athletic performance decreased with age, without significant differences between the study times (time for age interaction effect: β ± SE: −1.83 ± 2.43; *p*-value = 0.45). 

Considering the BAP level in Model B, antioxidant potential was directly associated with Yo-Yo test performance (β ± SE: 0.04 ± 0.01, *p*-value < 0.001), independently of age, season time, and cortisol levels. In the second-order analysis (Table 2, Model C), again, a direct statistically significant association could be reported for BAP (β ± SE: 0.09 ± 0.002, *p*-value < 0.001); no multiplicative effect for the interaction between age and BAP could be shown, but on the contrary, a statistically significant interaction between BAP and the time of the study (β ± SE: −0.07 ± 0.02, *p*-value < 0.001) was observed, meaning that, in the mid-season period compared with the pre-season period, a higher performance level for lower BAP levels was observed. Indeed, in the mid-season period, with a decrease of one unit in BAP, the Yo-Yo test performance was reduced by almost 7 cm. Interestingly, age was still inversely and significantly associated with the Yo-Yo test (β ± SE: −12.02 ± 2.82, *p* < 0.001), meaning that a decrease in performance of about 12 m is present for each year of age. Moreover, the time effect was not more statistically significant (β ± SE: −28.64 ± 51) since almost all of the variance was attributable to the BAP for time interaction. When the oxidant potential (U CARR) was considered, this marker was inversely associated with variation in performance in the Yo-Yo test (β ± SE: = −0.33 ± 0.10; *p*-value = 0.001), independently of age, time, and cortisol serum levels. Looking at the second-order analysis, a significant interaction between U CARR and the time of study can be appreciated (β ± SE: −1.09 ± 0.18, *p*-value < 0.001), as well a significant interaction between U CARR and age (β ± SE: −0.05 ± 0.02, *p*-value = 0.02), independently of cortisol serum levels. This means that the older subjects with higher U CARR values at follow-up had worse performance compared to those of younger age and lower U CARR values. The reduction in performance for every U CARR unit and every year of age was about 0.05 m. Moreover, for the same U CARR value from the pre-season to mid-season period, the Yo-Yo test performance was reduced by almost 1.09 m. Finally, in Model B, when the BAP/U CARR ratio, age, time of study, and cortisol serum levels were considered, again, an inverse association was found for age and the variation in performance in the Yo-Yo test (β ± SE: −11.72 ± 2.85, *p*-value < 0.001); the BAP/U CARR ratio was directly associated with variation in performance in the Yo-Yo test (β ± SE: 13.26 ± 2.22; *p*-value < 0.001). Looking at the second-order analysis, a statistically significant interaction for the BAP/U CARR ratio and the time of the study (β ± SE: −34.78 ± 3.79, *p*-value < 0.001) was found, whereas a borderline statistically significant multiplicative effect for ratio and age (β± SE: 0.81 ± 0.41, *p*-value = 0.06) was present. This means that from the pre-season period to the mid-season period, subjects with a higher BAP/U CARR ratio (i.e., with a relative prevalence of antioxidant potential in OS) performed better in comparison to the subjects with a lower BAP/U CARR ratio. In other words, for one unit of the ratio, the subjects with higher OS showed a mean reduction in performance of about 35 m.

## 4. Discussion

The findings of our research can be summarized as follows: (1) in the central part of the competitive season, performance in the Yo-Yo test increased significantly (about 20% in comparison to the pre-season period); (2) performance was significantly reduced according to age both in the pre-season period and in the mid-season period; (3) the changes in the blood levels of Vit D and folic acid, as well as those of testosterone and cortisol, did not show any relationship with athletic performance; (4) FR production as well that of antioxidant substances increased in the mid-season period after several training sessions and matches; (5) there was an inverse relationship between OS and improvement in performance, as higher levels of OS are associated with lower levels of performance in the Yo-Yo test; and (6) there was an inverse statistically significant multiplicative effect for the interaction between time for the BAP/U CARR ratio and the Yo-Yo test. 

### 4.1. Performance, Hormones, and Vitamins

The increase of about 20% in performance after training is in agreement with the values reported by other authors [29,30,31,34]. A significant decline in performance was observed in relation to age, regardless of biochemical markers. Indeed, a decline in several functions (the heart, lungs, muscles, metabolism, etc.) may occur with age; however, in a speculative way, it may be assumed that older athletes near the end of their career were less motivated in performing the test. According to previous studies [2,11,35], we found that the blood levels of Vit D and testosterone were higher in the summer. However, such a difference did not prove to be statistically significant in our research, probably, due to the large dispersion of values. Cortisol levels were higher during the winter, and hematologic biomarkers (iron, vitamin B12, and folic acid) did not show significant changes in the two periods of our study. These observations are not surprising. Indeed, it is well known that Vit D is mainly synthesized by the skin when exposed to ultraviolet B radiation and that the warm season is characterized by a more uninhibited lifestyle and sexual excitement, which enhances testosterone production. On the contrary, the higher cortisol levels in winter, at the heart of the competitive season, could be due to the harsh weather conditions and mainly due to the physical and psychological stress related to training and frequent competitions. However, it must be emphasized, as a new finding, that neither the first nor the latter did show any relationship with the rate of improvement in performance in the Yo-Yo test. 

### 4.2. Performance and Oxidative Stress

The more relevant findings of our study are those concerning the changes in the parameters of OS during the season and their relationships with the rate of improvement in athletic performance after training. The increase in FR production as well that of antioxidant substances in the mid-season period after several training sessions and matches has been investigated in other studies, with partly discrepant results. Zivkovic et al. evaluated the levels of some markers of OS in the pre-season period and after six months of training in young soccer players (12–13 years old) [36]. These authors found that the levels of the index of lipid peroxidation (measured as TBARS) and nitrites significantly increased, while the superoxide anion radical and the hydrogen peroxide remained unchanged [36]. On the other hand, superoxide dismutase and catalase activity increased, while that of reduced glutathione (GSH) decreased. Their conclusion was that a carefully prepared training program could strengthen most components of the antioxidant defense systems and that it did not promote OS except for lipid peroxidation. Le Moal et al. evaluated variations in pro-/antioxidant status throughout a whole season and their relationships with training load in elite professional soccer players of the French league (n = 19, 18.3 ± 0.6 years) [37]. The GSH/GSSG ratio, a marker of antioxidant potential, increased significantly between the season periods, showing a significant correlation with the mean of the training loads. Ponce-Gonzalez et al. evaluated antioxidant blood markers in four soccer teams in the first week of the championship season (pre-season) and after 18 weeks in the mid-season period, recording the performance of the soccer players according to the official classification ranking [23]. Regression analyses showed that a higher antioxidant capacity was present in the players of the best performing team, both in the mid-season period and at the end of the season. Sadowska-Krępa assessed changes in antioxidant status in American football players and soccer players over a training macrocycle [24]. The activity levels of superoxide dismutase, glutathione peroxidase, catalase, glutathione reductase, creatine kinase, and lactate dehydrogenase, as well as the concentrations of non-enzymatic antioxidants (uric acid and glutathione) and the levels of malondialdehyde, were evaluated at the beginning of the three periods (preparatory, competition, and transition) making up the training macrocycle [24]. The study revealed moderate improvements in the blood antioxidant status of the athletes at the end of the preparatory period (in the early competition period), which was shown to be associated with their training loads. Considering the differences in the study protocols, the age of the soccer players, the time of observation, the different training programs, and the biochemical parameters evaluated, it is difficult to compare the results of these studies with each other and with ours. However, it is evident that the increase in FR production as well as the simultaneous increase in BAP after training sessions and matches is a common finding to all the studies. Finally, to our knowledge, only one study has investigated the relationship between OS and athletic performance assessed by means of standardized tests [26]. This study was performed by Silva et al., who evaluated the relationship between the levels of OS, testosterone, and cortisol and the results of the following tests: 5 and 30 m sprints, countermovement jump, maximal isokinetic knee extension, and knee flexion strength in 14 professional soccer players at four different times in the season. Significant associations were observed between the match-accumulated time (minutes played by each athlete during the competition period), test performance, and hormonal and redox parameters [26]. Unfortunately, despite the similar experimental design, the results of this study cannot be fully compared with those of the present research, given the differences in the performance tests employed and the calculation methods, which in our study aimed to evaluate whether OS could be predictive of or associated with the rate of change in performance before and after intensive training. 

The new findings of our research are that the higher increase in performance (calculated as the difference in meters run in the Yo-Yo test between the pre-season and mid-season periods) is significantly associated with a minor increase in OS. In detail, higher increases in performance were associated with reduced production of FRs; moreover, a significant interaction between age and OS was demonstrated, meaning that older subjects, who also had higher U CARR values, at follow-up showed worse performance when compared to those of younger age with lower U CARR values. BPA was shown to be increased in the mid-season period and directly associated with Yo-Yo test performance. Interestingly, no multiplicative effect for the interaction between age and BAP could be shown, but on the contrary, a statistically significant interaction between BAP and the time of the study was observed: meaning that when comparing the data from the mid-season and pre-season periods, a higher level of performance for similar BAP levels was found in the first one. Finally, the BAP/U CARR ratio was also directly associated with the variation in performance in the Yo-Yo test; in other words, the subjects with a higher BAP/U CARR ratio (i.e., with a relative prevalence of antioxidant potential in FR production) performed better (about 35 m for one unit of the ratio) from the pre-season period to the mid-season period in comparison with the subjects with a lower BAP/U CARR ratio. The results of the present study are explained by basic research. Excessive FRs produced by strenuous exercise may cause fatigue (and therefore interruption of the Yo-Yo test) by reducing the release of skeletal muscle sarcoplasmic reticulum Ca^2+^ and/or the sensitivity of myofibrils Ca^2+^ [38]. Moreover, endogenous and exogenous ROS can also promote the occurrence of exercise fatigue by destroying mitochondrial function and inhibiting aerobic metabolism [39]. In addition, during exercise, the increased energy metabolism rate leads to excessive mitochondrial ROS production in fatigued skeletal muscle cells, which further leads to the occurrence of the oxidative modification of lipids, proteins, and DNA. When the individual production of antioxidant substances is adequate, these molecules may partially offset the negative effects of FRs, restoring the release of Ca^2+^ in the sarcoplasmic reticulum, thus delaying the occurrence of muscular fatigue [40]. The reasons why athletes exhibit a different individual pattern in the production of pro- and antioxidant substances under strenuous exercise remain unknown [41,42]. Excluding the possibility that individual behavior could be related to significant differences in diet, lifestyle habits (such as smoking and alcohol use), exposure to environmental toxins, and, above all, different training loads, as they were almost the same for all subjects, the intervention of genetic factors may be supposed. With it being impossible to modify the genetic component, in order to improve the performance of the subjects with an unfavorable oxidative system, two ways can be considered. Firstly, a proper individual training schedule (frequency, intensity, and characteristics of exercise) must be considered. Indeed, short exercise sessions produce fewer FRs than long ones, and they may also contribute to building up optimal defenses against FR damage, particularly by “learning” how to protect muscle tissue [25]. In addition, supplementation with antioxidant substances can be taken into consideration, even though their use must be subject to caution and limited only to those who do not seem to benefit from a proper change in training schedule [43,44]. Indeed, high doses of antioxidants taken over long periods of time can impair the antioxidant adaptive response of specific circulating cell types by avoiding the optimal expression of genes coding for antioxidant enzymes [43,44]. In our opinion, adequate intake of vitamins and minerals through a varied and balanced diet remains the best approach to maintain optimal antioxidant status in exercising individuals.

### 4.3. Limitations

Some limitations of the present research must be acknowledged. First, the assay of biological substances under study was performed only before the Yo-Yo test and not after its completion. Indeed, it could be useful to assess to what extent strenuous exercise may have modified the production of pro- and antioxidant substances in our athletes; unfortunately, this measure was not included in the routinary assessment protocol. Secondly, the validity of our results is limited to the Yo-Yo test since it is well known that exercises of different intensity and duration result in different levels of biological stress [45]. Therefore, the use of a different test of performance might show different results. However, it must be underlined that the Yo-Yo test is characterized by high-intensity exercise, with sudden accelerations and decelerations, which mimic the behavior of athletes in the field, and therefore, it is considered a reliable and valid measurement instrument of match-related performance in professional soccer athletes [29,30,31]. In the official sport seasons, the soccer players did not report the consumption of any spontaneous supplements or drugs, but omission or recall bias could not be excluded, and this situation could potentially influence the test performance results. We did not take into account the different distances run by the players on the field during the matches, the speed, the sprints, or the accelerations and decelerations, which vary according to their position and role on the field. Lastly the observational retrospective study design did not enable us to disentangle the causal relationship direction between OS and athletes’ performance, which could be better clarified by larger clinical trials.

## 5. Conclusions

Training sessions and competitions induce physiological changes in soccer players, who require specific strategies to optimize their efficiency, monitor overload, and prevent the risk of injuries. The main finding of this observational study is that OS is associated with a gain in physical performance between pre-season and mid-season periods. Indeed, the athletes with higher levels of FRs and/or a lower production of antioxidant substances exhibited a reduced gain in physical performance after training. In conclusion, it could be useful for coaches to monitor the measures of OS in their players as a possible harbinger of future development of overtraining syndrome.

## Figures and Tables

**Figure 1 antioxidants-12-01470-f001:**
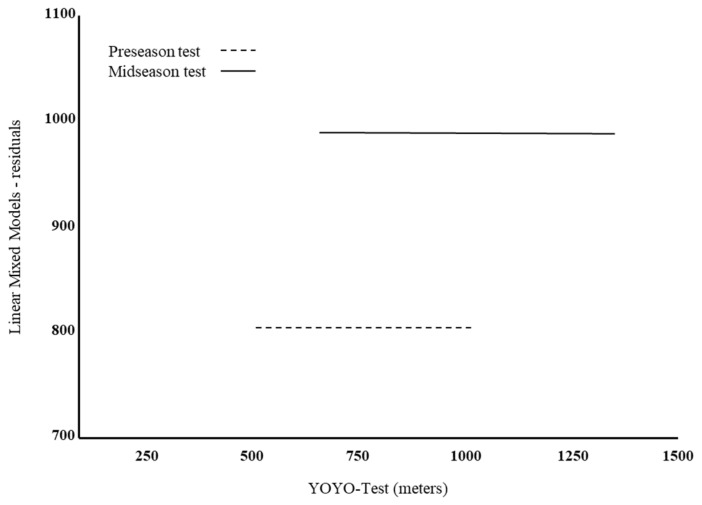
Linear mixed models analysis residuals versus the YoYo test according to time. The dashed line represents the pre-season test; the continuous line represents the mid-season test. The starting point of the lines refers to the athletes who had the worst performance, whereas the end point refers to the athletes who had the best performance.

**Figure 2 antioxidants-12-01470-f002:**
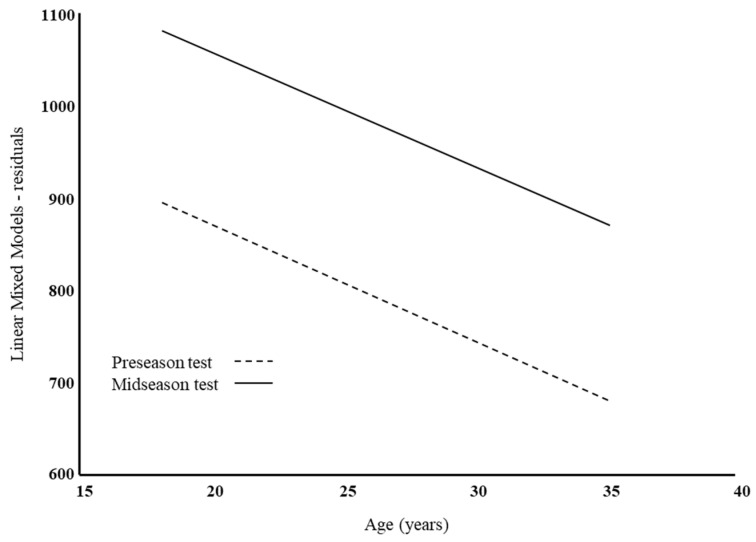
Linear mixed model analysis residuals versus age according to the time of the study. The cortisol serum levels were adjusted.

**Table 1 antioxidants-12-01470-t001:** Anthropometric measures, the Yo-Yo performance test, and laboratory parameters in the pre-season and mid-season periods (quantile regression was applied to assess differences between the times).

	Pre-Season	Mid-Season	*p*-Value
Number	82	82	
Age (year)	25.3 ± 4.2	25.3 ± 4.2	
Weight (kg)	75.1 ± 6.2	73.7 ± 4.8	0.42
Height (cm)	176.8 ± 5.9	176.8 ± 6	0.99
BMI (kg/m^2^)	23.9 ± 1	23.5 ± 0.8	0.36
Yo-Yo test (m)	799.8 ± 116.4	984.1 ± 153.6	<0.001 *
Hemoglobin (g/dL)	15 ± 0.6	14.9 ± 0.5	0.07
Iron (µg/dL)	97.6 ± 25.3	96 ± 21.9	0.76
Vitamin B12 (pg/mL)	667.6 ± 174.8	655.1 ± 166.1	0.98
Folic Acid (ng/mL)	82.2 ± 24.3	79.4 ± 20.6	0.83
Vitamin D (ng/dL)	29.1 ± 6.6	26 ± 6.8	0.16
Testosterone (ng/mL)	86.9 ± 24	77.6 ± 24	0.17
Cortisol (µg/dL)	117.7 ± 44.4	177.4 ± 77.7	0.001 *
Creatin kinase (µg/L)	231.1 ± 106.5	233.8 ± 116.4	0.70
Oxidative stress (U CARR)	242.5 ± 74.2	287.7 ± 56.6	0.02 *
BAP (μmol/L)	2233.8 ± 772.7	2443.2 ± 467	<0.001 *
BAP/U CARR ratio	9.6 ± 2.9	8.7 ± 2.2	0.58

BMI = body mass index; U CARR = carratine units; BAP = blood antioxidant potential. * highlights statistically significant results for differences between the times.

**Table 2 antioxidants-12-01470-t002:** Linear mixed model: Analysis of Yo-Yo performance test variation in the two periods of the study according to age. Model A: unconditional means model; Model B: unconditional growth model; Model C: assessed the BAP effect; Model D: assessed the U CARR effect; Model E: assessed the BAP/U CARR ratio effect. Models C, D, and E were adjusted for cortisol serum level.

		Model A	Model B	Model C	Model D	Model E
Initial status						
Intercept	γ_00_	893.4 ± 13.9 ***	985.6 ± 14.8 ***	1105 ± 94 ***	1303 ± 154 ***	1131 ± 128 ***
Age	γ_01_			−12.02 ± 2.8 ***	−0.2 ± 5.7	−18.7 ± 4.5 ***
BAP	γ_02_			0.09 ± 0.02 ***		
UCARR	γ_03_				0.1 ± 0.5	
Ratio	γ_04_					15.4 ± 11.05
Age*U CARR	γ_01*03_				−0.05 ± 0.02 *	
Age*Ratio	γ_01*04_					0.8 ± 0.4
Rate of change						
Intercept	γ_10_		−184.3 ± 10.4 ***	−28.6 ± 51.7	502.2 ± 8.2 ***	123.3 ± 37.2 **
Time*BAP	γ_12_			−0.07 ± 0.02 ***		
Time*U CARR	γ_13_				−1.09 ± 0.1 ***	
Time*Ratio	γ_14_					−34.7 ± 3.7 ***
Within person	δ^2^_e_	21,495 ± 3336 ***	4505 ± 699 ***	3559 ± 553 ***	2640 ± 412 ***	1563 ± 244 ***
In initial status	δ^2^_0_	5313 ± 3000 *	13,808 ± 2517 ***	10,259 ± 1890 ***	10,155 ± 1805 ***	10,089 ± 1704 ***
	AIC	2166	2039	2003	1974	1927

U CARR = carratine units; BAP = blood antioxidant potential. γ_00_ = intercept of the average trajectory; γ_01_ = intercept of the trajectory for Age; γ_02_ = intercept of the trajectory for BAP; γ_03_ = intercept of the trajectory for U CARR; γ_04_ = intercept of the trajectory for BAP/U CARR; γ_01*03_ = intercept of the trajectory for Age*U CARR; γ_01*04_ = intercept of the trajectory for Age*BAP/U CARR; γ_10_ = slope of the average trajectory; γ_12_ = slope of the average trajectory for time*BAP; γ_13_ = slope of the average trajectory for time*U CARR; γ_14_ = slope of the average trajectory for time*BAP/U CARR; δ^2^_e_ = within-person variance components; δ^2^_0_ = in initial status variance components. * *p* < 0.05; ** *p* < 0.01; *** *p* < 0.001.

## Data Availability

The datasets used and/or analyzed during the current study are available from the corresponding author upon reasonable request.

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
