# Peer review of "Oxidative Stress and Performance after Training in Professional Soccer (European Football) Players"

_antioxidants, 2023, doi:10.3390/antiox12071470_

Round 1

Reviewer 1 Report (Previous Reviewer 1)

The ethics committee approval was received and all other issues were corrected.

Author Response

The authors thank the Reviewers for their support in writing the paper and for the time they donated.

Reviewer 2 Report (Previous Reviewer 2)

The authors have revised their manuscript and have obtained the necessary ethics approval for the study. I have a few additional comments / suggestions for revision:

Table 1: Results are presented from pre-season to mid-season, but it is stated in the title and footnote to the table that these are “between group” comparisons”. Are these not “repeated measures” comparisons? I think this should be re-worded to indicate these are comparisons over time and not between groups.

Table 2: It is unclear what *, **, and *** represent. It is unclear what “In initial status” means. “Unconditional Means Model” and “Unconditional Growth Model” – Please check that “unconditional” is the correct word to use here.

Line 430: Change “player” to “players”

Line 430-431: Change “assumption” to “consumption”

The English requires minor revision

Author Response

Table 1: Results are presented from pre-season to mid-season, but it is stated in the title and footnote to the table that these are “between group” comparisons”. Are these not “repeated measures” comparisons? I think this should be re-worded to indicate these are comparisons over time and not between groups.

  • Your observation is correct, and we emended the text accordingly.

Table 2: It is unclear what *, **, and *** represent. It is unclear what “In initial status” means. “Unconditional Means Model” and “Unconditional Growth Model” – Please check that “unconditional” is the correct word to use here.

  • We are sure about the terminology used to identify the different models, all the “statistical-term” used in this manuscript and all the approaches derived from JD Singer courses and from her publications; we insert a new Reference in the statistical section that recall the methodology suggested by Singer in Longitudinal analysis.
  • We emended the text accordingly to this reviewer suggestion.

Line 430: Change “player” to “players”

Line 430-431: Change “assumption” to “consumption”

  • We emended the text accordingly to this reviewer suggestions.

Reviewer 3 Report (Previous Reviewer 4)

The article has been corrected, especially the ethical aspect has been clarified and supplemented. The article is correct in its current form. The authors have corrected all the issues that were previously an obstacle to a positive opinion of the article. I recommend accepting the article in its current version.

Author Response

The authors thank the Reviewers for their support in writing the paper and for the time they donated.

This manuscript is a resubmission of an earlier submission. The following is a list of the peer review reports and author responses from that submission.

Round 1

Reviewer 1 Report

The manuscript of Michele Abate et al. entitled „Oxidative Stress and Performance After Training in Profes-2 sional Soccer Players” highlights the significance of oxydative stress and other parameters in the performance of football players. The manuscript is generally well-written, but there are some inconsistencies, that should be cleared before pubication.

Major:

1. Oxidative stress should be defined, or at least mentioned in the following sentence: „The measurement is expressed in U Carr (Carratine Units)…”. Rather: „The measurement of OS is expressed in U Carr...”

2. What was the reason of measuring some players 3 times, other players only 1-2 times. Was any comparison made between the 3 results of the 14 persons? Were the data of players with only 1 result analyzed in their own? Was there any difference between the result of the 2 populations?

3. Table 1.: if the 2 tests were carried out in august and in February, why there is no change in the age between the 2 tests? One would expect 0.5 year higher in the 2nd case. Additionally, it should be indicated in the table e.g. with *, which are the significant results.

4. Same issue like in point 3. How was the age taken into consideration during the linear mixed model?

5. In the beginning of the discussion, where the most important results are summarized in 5 points, BAP / U CARR ratio should be also mentioned.

Minor:

6. It should be also mentioned, that soccer means the European football, because most of the potential readers know rather this name.

7. Why are cortisol and testosteron abbreviated? Both are short words.

8. Abstact.: be-tween, - is not needed

9. Measures: -20oC should be rather -20 °C

10. Discussion: I think there is no reason to write „authors” bold.

Author Response

We appreciate a lot the reviewers’ suggestions, and we have to thank them for the help in pointing out the criticisms of our manuscript.

Reviewer #1:

  1. Oxidative stress should be defined, or at least mentioned in the following sentence: „The measurement is expressed in U Carr (Carratine Units)…”. Rather: „The measurement of OS is expressed in U Carr...”

We agree with this reviewer, and accordingly we change the text, that now appear as:

The measurement of OS is expressed in U Carr (Carratine Units), where 1 U Carr corresponds to 0.08 mg/dl H2O2. The suggested FRs upper limit is >300 U Carr. A value >500 U Carr is considered expression of severe oxidative stress. (lines 133-136).

  1. What was the reason of measuring some players 3 times, other players only 1-2 times. Was any comparison made between the 3 results of the 14 persons? Were the data of players with only 1 result analyzed in their own? Was there any difference between the result of the 2 populations?

We thank the reviewer for the suggestion. The data used in this analysis are derived from a professional Italian soccer team, and every year the team could change; therefore, some players could be part of the team in all the times of the study or alternatively they were part of the team for only one year. Moreover, to account for this statistical problem we use LMM that in contrast to Anova for repeated measures, do not need that all subjects had all times follow-up. Therefore, this analytic approach was not biased by the numerosity of the sample in the single time (see Judith Singer ALDA Applied Longitudinal Data Analysis). No comparisons were done between the three clusters at the univariate analysis level, but in the LMM, when we had to specify the identity link for repeated measures, was also considered the nested factor for the year of the physical assessment.

  1. Table 1.: if the 2 tests were carried out in august and in February, why there is no change in the age between the 2 tests? One would expect 0.5 year higher in the 2nd case. Additionally, it should be indicated in the table e.g. with *, which are the significant results.

We agree with this reviewer; obviously after 6 months, all subjects were 6 months older, this is a systematic error; therefore, it could not influence any results, and is constant for all the subjects enrolled in the study. Moreover 5-6 months of age change are considered in the time-effect between the two follow-ups; age could be considered as a birth-cohort effect, instead a chronological age. Therefore, if we consider the change of time in both variables, age and time we could bias the time effect-estimation. Lastly, we did not understand the second part of the suggestion, we have clearly stated and reported in the Table 1, which are the statistically significant differences between the two times of the study, last column named: p-value.

  1. Same issue like in point 3. How was the age taken into consideration during the linear mixed model?

We appreciate a lot the suggestion of this reviewer, and accordingly we stated in the statistical section that age was forced in model C, D and E. To better explain, age was considered in the model as a continuous variable, rounded to the nearest year. Anyway, we change the text to better elucidate the use of age and cortisol as potentially confounders, also considering that at the univariate analysis differences between the two times of the study, for both variables were statistically significant.

“each parameter was considered as the independent variable (fixed effects), in different models; moreover age, and Cortisol serum level were considered in the models as potentially confounders. (Lines 175-176)

  1. In the beginning of the discussion, where the most important results are summarized in 5 points, BAP / U CARR ratio should be also mentioned.

We had to thank this Reviewer for the suggestion, and accordingly we consider in the beginning of the discussion also the result for the Ratio.

… 6) there is an inverse statistically significant multiplicative effect for the interaction be-tween time for BAP/U CARR ratio, and YoYo test. (Lines 291-293).

Minor:

  1. It should be also mentioned, that soccer means the European football, because most of the potential readers know rather this name.
  2. Why are cortisol and testosteron abbreviated? Both are short words.
  3. Abstact.: be-tween, - is not needed
  4. Measures: -20oC should be rather -20 °C
  5. Discussion: I think there is no reason to write „authors” bold.

We apologize for all the mistakes, and or typos, highlighted by this Reviewer, we accordingly amended.

Reviewer 2 Report

This study is on an interesting topic; however, there are two major limitations / concerns: 1) Ethics approval is required for this study...it is stated this study was not approved by an ethics board; 2) I am uncertain whether you can combine within-participant data over multiple seasons (multiple pairs of tests from the same participants) along with tests from single participants in the same statistical model.

Specific comments:

Abstract, line 25: “as well the blood antioxidant potential.” Change to “as well as the blood antioxidant potential.”

Line 25: “An indirect relationship be-tween oxidative stress and the improvement in performance was observed, lower levels of oxidative stress being associated to higher levels of performance in the YoYo test.” – In the abstract, could you provide more specific information on this result, with numerical results and/or statistical results?

Line 77: “at our best knowledge” – change to “to our best knowledge”

Line 91: “Institutional ethics committee approval was not required because the measures were part of the usual athletic assessment, and because the observational nature of the study” – I would suggest that institutional ethics approval is required in this instance since you are using data collected on human participants for research purposes. The study should have been reviewed for assessment of risks and to ensure proper procedures are taken to protect confidentiality of research participants.

Line 108: “A clinical control was regularly performed.” – It is unclear what is meant by this statement

Line 122: Superscript the “2” in “kg/m2”

Line 122: “Therefore” is not the correct word to use here.

Line 127: “-20oC” – correction is needed here for the degrees symbol

Line 134: “H2O2” – some subscripting is required here

Line 148: Please provide the manufacturer, city, etc. for these ELISA kits

Line 195: “Forty-two athletes were studied during three agonistic seasons (2018-2021): 14 performed the tests in all the three seasons, 12 only in two seasons, and 16 only in one season. On the whole, 82 paired tests were evaluated.” I am not sure if you can combine multiple paired tests from the same individual in this analysis. I would think that each individual should be only assessed across one set of repeated tests. I am not sure you can combine between-subjects and within-subjects data like this.

Line 208: Delete “Tables may have a footer.”

Table 1: I suggest you define all abbreviations used in the table in a footnote to the table.

Line 210: “in mt” – it is unclear what is meant by this.

Table 2: It is unclear what is meant by “unconditional” here. Did you mean “unadjusted”?

Table 2: Please define all abbreviations in a footnote to the table.

Line 252: Again, “mt” is not an appropriate abbreviation here

There are many abbreviations in the manuscript, making it difficult to read.

Line 288: “authors” should not be bolded here

Lines 323-324: “performance of the soccers” – some re-wording is needed here.

Lines 371 and 372: “Ca2+” – superscripting is required here

Line 399: “the essay of biological substances” – do you mean “assay” here instead of “essay”?

Author Response

This study is on an interesting topic; however, there are two major limitations / concerns:

1) Ethics approval is required for this study...it is stated this study was not approved by an ethics board;

We partially agree with this reviewer, and we explain our point of view:

  1. First of all, this is an observational retrospective study, therefore for the EC was not possible to suggest changes to the protocol;
  2. Data are anonymized in the DB, in a way that requires excessive efforts to be re-identified, then the data could not be related to a specific person anymore. Only first authors (MA) could disentangle the identification code; for example, who performed the statistical analysis was blinded about the identity of the participants;
  3. All results produced by our analysis were reported as aggregate, therefore could not be identified subjects involved in the study;
  4. All the authors certified non-involvement of any minors, vulnerable groups, or sensitive topics, and that they were transparent about study purpose, confidentiality, privacy, anonymity, of the subjects involved in this study;
  5. According to: article 26 of the REGULATION (EU) 2016/679 OF THE EUROPEAN PARLIAMENT AND OF THE COUNCIL of 27 April 2016 on the protection of natural persons with regard to the processing of personal data and on the free movement of such data, and repealing Directive 95/46/EC (General Data Protection Regulation), that sound as: “The principles of data protection should therefore not apply to anonymous information, namely information which does not relate to an identified or identifiable natural person or to personal data rendered anonymous in such a manner that the data subject is not or no longer identifiable. This Regulation does not therefore concern the processing of such anonymous information, including for statistical or research purposes.”
  6. According to article 44 (same reference of point E): Processing should be lawful where it is necessary in the context of a contract or the intention to enter into a contract;
  7. According to article 50 (same reference of point E): omissis …” Further processing for archiving purposes in the public interest, scientific or historical research purposes or statistical purposes should be considered to be compatible lawful processing operations.”
  8. In Italy by law, for every professional sport, athletes must undergo a complete assessment at least in two times during the sport season, and the Sport-societies are responsible for the execution, the referent medical doctor was the person in charge to manage data privacy;
  9. At the moment of the contract signature the athletes accept this condition;
  10. All the athletes accepted to donate their lab-results and to be part of this study, with a second specific informed consent (previously sent to the Assistant Editor);
  11. Lastly the study design is an observational retrospective study; therefore, all the data that we obtained and used could not be considered an active experiment; all data are routinely collected, stored in a database by the lab who performed assays, and determined by the Italian Law.

2) I am uncertain whether you can combine within-participant data over multiple seasons (multiple pairs of tests from the same participants) along with tests from single participants in the same statistical model.

Due to specific data collection design, unbalanced repeated measures, we applied Linear Mixed Models, a statistical approach that could manage within-between-subjects variance assessment, also for unbalanced data between times of the study. For more details see also Judith Singer, Applied Longitudinal data analysis, also referenced in the bibliography.

Briefly LMM, is more powerful compared to ANOVA for repeated measures, because the procedure can manage different follow-up assessment numbers, between enrolled subjects; moreover, LMM allow to model analysis according to your hypothesis. In other word if you are interested in a second order interaction for example “time for BAP” age adjusted, it is not necessary to consider in the model also the second order interaction between “time for Age”, as required by ANOVA.

Unconditional means model and unconditional growth model are the nomenclature suggested by Judith Singer for LMM, and in a more formal paradigm:

For a random variable γt, the unconditional mean is simply the expected value, E(γt). In contrast, the conditional mean of γt is the expected value of γt given a conditioning set of variables, Ωt. A conditional mean model specifies a functional form for E(γt∣Ωt).

Specific comments:

  • Abstract, line 25: “as well the blood antioxidant potential.” Change to “as well as the blood antioxidant potential.”
  • The abstract’s text was changed;
  • Line 25: “An indirect relationship be-tween oxidative stress and the improvement in performance was observed, lower levels of oxidative stress being associated to higher levels of performance in the YoYo test.” – In the abstract, could you provide more specific information on this result, with numerical results and/or statistical results?
  • Accordingly, to your suggestion we reported in the abstract the LMM estimates for UCAR;
  • Line 77: “at our best knowledge” – change to “to our best knowledge”
  • Accordingly, to your suggestion we change the text;
  • Line 91: “Institutional ethics committee approval was not required because the measures were part of the usual athletic assessment, and because the observational nature of the study” – I would suggest that institutional ethics approval is required in this instance since you are using data collected on human participants for research purposes. The study should have been reviewed for assessment of risks and to ensure proper procedures are taken to protect confidentiality of research participants.
  • See our rebuttal introduction;
  • Line 108: “A clinical control was regularly performed.” – It is unclear what is meant by this statement:
  • This part of the text was erased
  • Line 122: Superscript the “2” in “kg/m2”
  • Accordingly we change the text;
  • Line 122: “Therefore” is not the correct word to use here.
  • Therefore was erased from the text
  • Line 127: “-20oC” – correction is needed here for the degrees symbol
  • Accordingly we made the correction
  • Line 134: “H2O2” – some subscripting is required here
  • Accordingly we made the correction

  • Line 148: Please provide the manufacturer, city, etc. for these ELISA kits
  • Accordingly we have emended the typos
  • Line 195: “Forty-two athletes were studied during three agonistic seasons (2018-2021): 14 performed the tests in all the three seasons, 12 only in two seasons, and 16 only in one season. On the whole, 82 paired tests were evaluated.” I am not sure if you can combine multiple paired tests from the same individual in this analysis. I would think that each individual should be only assessed across one set of repeated tests. I am not sure you can combine between-subjects and within-subjects data like this.
  • See introduction to our rebuttal;
  • Line 208: Delete “Tables may have a footer.”
  • We apology for the typo;
  • Table 1: I suggest you define all abbreviations used in the table in a footnote to the table.
  • Accordingly, to your suggestion we use the footnote to specify the abbreviations;
  • Line 210: “in mt” – it is unclear what is meant by this.
  • We check the manuscript and made the appropriate changes;
  • Table 2: It is unclear what is meant by “unconditional” here. Did you mean “unadjusted”?
  • See introduction of our rebuttal;
  • Table 2: Please define all abbreviations in a footnote to the table.
  • Accordingly, to your suggestion we use the footnote to specify the abbreviations;
  • Line 252: Again, “mt” is not an appropriate abbreviation here There are many abbreviations in the manuscript, making it difficult to read.
  • Accordingly, to your suggestion we made the appropriate changes
  • Line 288: “authors” should not be bolded here
  • Is a typo due to the template characteristic;
  • Lines 323-324: “performance of the soccers” – some re-wording is needed here.
  • We change in soccer-players
  • Lines 371 and 372: “Ca2+” – superscripting is required here
  • We have checked the text and we have made the appropriate changes.
  • Line 399: “the essay of biological substances” – do you mean “assay” here instead of “essay”
  • We have to thanks the reviewer for the suggestions and help in reviewing the text.

Reviewer 3 Report

This study investigates the the association between biological mediators and rate of performance improvement in soccer players after training.

Line 14: Change " associated to" to "associated with" and "in the performance" to "in performance"

Line 16: Change "second and at mid-point" to "second at the mid-point"

Line 20: Change "increase of meters ran in the Yo-Yo" to "increase in meters run in the Yo-Yo"

Line 21: Change "of the Linear Mixed effect Models" to "of a Linear Mixed effects Model"

Line 21: Delete "Results: On the whole," and change "82" to "Eighty two"

Line 22: Delete "As expected,"

Line 24: Change "vitamins and hormones" to vitamin and hormone"

Line 25: Change "as well" to "as did"

Line 26: Change "be-tween" to "between"

Line 27: Change "associated to" to "associated with" and "YoYo test" to "Yo-Yo test"

Line 28: Change "to suggest coaches some changes in the training" to "to suggest to coaches that they make some changes in the training"

Line 29: Change "adequate to each" to "adequate for each"

Line 30: Delete "a"

Line 43: Change "risk for" to risk of"

Line 46: Change "been considered harbinger" to "been considered a harbinger"

Line 47: Change "the free radical (FRs) production" to "production of free radicals (FR) increases significantly" and change FRs to FR throughout the manuscript.

Line 77: Changer "...at our best knowledge..." to "...to the best of our knowledge..."

Line 78: Change "relationships" to "relationship"

Line 89: I strongly disagree with your reason for not registering the study with an ethics committee. If athletes were asked to provide written consent, were fasted overnight before assessment, which included blood samples, even if the volunteers are not performing any tasks outside their usual activities, an ethics committee must be aware of the study and it should, therefore be registered. 

Line 91: Change to "...and gave written consent to participate in the study."

Lines 100 - 104: This detail of the season schedule is irrelevant, please delete it.

Line 107: Change "into" to "in"

Lines 112 - 116: How was player diet monitored?

Line 153: What constitutes a standard breakfast? You need to specify and provide a reference.

Lines 168, 249: Change "compared to" to "compared with"

Line 172 & 353: Change "ran" to "run"

Line 193: Change "2-sided" to "2-tailed"

Line 242: Change to "Considering the BAP-level in Model B, the antioxidant potential..."

Line 249: Change to "...compared with pre-season, a higher performance level for lower BAP levels was observed."

Line 250: Change to "...a decrease of one unit..."

Line 251: Change "...reduced of..." to "...reduced by..."

Line 253 & 344: Change "12 mt" to "12 m"

Line 2567: Change "...performance at..." to "...performance in..."

The first few sentences of the Discussion just restate the results and can be deleted up to line 287.

Author Response

We apologize for all the typos, founded by this Reviewer, we accordingly amended the text, and contextually we have to thank the reviewer for the time spent to help us in the revision of the manuscript.

Line 14: Change " associated to" to "associated with" and "in the performance" to "in performance"

Line 16: Change "second and at mid-point" to "second at the mid-point"

Line 20: Change "increase of meters ran in the Yo-Yo" to "increase in meters run in the Yo-Yo"

Line 21: Change "of the Linear Mixed effect Models" to "of a Linear Mixed effects Model"

Line 21: Delete "Results: On the whole," and change "82" to "Eighty-two"

Line 22: Delete "As expected,"

Line 24: Change "vitamins and hormones" to vitamin and hormone"

Line 25: Change "as well" to "as did"

Line 26: Change "be-tween" to "between"

Line 27: Change "associated to" to "associated with" and "YoYo test" to "Yo-Yo test"

Line 28: Change "to suggest coaches some changes in the training" to "to suggest to coaches that they make some changes in the training"

Line 29: Change "adequate to each" to "adequate for each"

Line 30: Delete "a"

Line 43: Change "risk for" to risk of"

Line 46: Change "been considered harbinger" to "been considered a harbinger"

Line 47: Change "the free radical (FRs) production" to "production of free radicals (FR) increases significantly" and change FRs to FR throughout the manuscript.

Line 77: Changer "...at our best knowledge..." to "...to the best of our knowledge..."

Line 78: Change "relationships" to "relationship"

Line 89: I strongly disagree with your reason for not registering the study with an ethics committee. If athletes were asked to provide written consent, were fasted overnight before assessment, which included blood samples, even if the volunteers are not performing any tasks outside their usual activities, an ethics committee must be aware of the study and it should, therefore be registered. 

We appreciate this Reviewer suggestion/observation, but we would like to explain the reasons’ background:

  1. First of all, this is an observational retrospective study, therefore for the EC was not possible to suggest changes to the protocol;
  2. Data are anonymized in the DB, in a way that requires excessive efforts to be re-identified than the data could not be related to a specific person anymore. Only first authors (MA) could disentangle the identification code; for example, who performed the statistical analysis was blinded about the identity of the participants;
  3. All results produced by our analysis were reported as aggregate, therefore could not be identified subjects involved in the study;
  4. All the authors certified non-involvement of any minors, vulnerable groups, or sensitive topics, and that they were transparent about study purpose, confidentiality, privacy, anonymity, of the subjects involved in this study;
  5. According to: article 26 of the REGULATION (EU) 2016/679 OF THE EUROPEAN PARLIAMENT AND OF THE COUNCIL of 27 April 2016 on the protection of natural persons with regard to the processing of personal data and on the free movement of such data, and repealing Directive 95/46/EC (General Data Protection Regulation), that sound as: “The principles of data protection should therefore not apply to anonymous information, namely information which does not relate to an identified or identifiable natural person or to personal data rendered anonymous in such a manner that the data subject is not or no longer identifiable. This Regulation does not therefore concern the processing of such anonymous information, including for statistical or research purposes.”
  6. According to article 44 (same reference of point E): Processing should be lawful where it is necessary in the context of a contract or the intention to enter into a contract;
  7. According to article 50 (same reference of point E): omissis …” Further processing for archiving purposes in the public interest, scientific or historical research purposes or statistical purposes should be considered to be compatible lawful processing operations.”
  8. In Italy by law, for every professional sport, athletes must undergo a complete assessment at least in two times during the sport season, and the Sport-societies are responsible for the execution;
  9. At the moment of the contract signature the athletes accept this condition;
  10. All the athletes accepted to donate their lab-results and to be part of this study, with a second specific informed consent (previously sent to the Assistant Editor);
  11. Lastly the study design is an observational retrospective study; therefore, all the data that we obtained and used could not be considered an active experiment; all data are routinely collected, stored in a database by the lab who performed assays, and determined by the Italian Law.

Line 91: Change to "...and gave written consent to participate in the study."

Lines 100 - 104: This detail of the season schedule is irrelevant, please delete it.

Line 107: Change "into" to "in"

Lines 112 - 116: How was player diet monitored?

  • We have reported in the text that: All players were followed actively by a dietician with a special interest in sport’s nutrition (lines 107)..

Line 153: What constitutes a standard breakfast? You need to specify and provide a reference.

Accordingly, to your suggestion we add the reference, but we did not specify the dietetic approach since it was personalized according to players’ needs.

Lines 168, 249: Change "compared to" to "compared with"

Line 172 & 353: Change "ran" to "run"

Line 193: Change "2-sided" to "2-tailed"

Line 242: Change to "Considering the BAP-level in Model B, the antioxidant potential..."

Line 249: Change to "...compared with pre-season, a higher performance level for lower BAP levels was observed."

Line 250: Change to "...a decrease of one unit..."

Line 251: Change "...reduced of..." to "...reduced by..."

Line 253 & 344: Change "12 mt" to "12 m"

Line 2567: Change "...performance at..." to "...performance in..."

The first few sentences of the Discussion just restate the results and can be deleted up to line 287.

  • In agreements with the suggestions made by Reviewer 1, we believe that such data summary could be useful; indeed, we believe that these few sentences can clearly highlight the important results of the study, introducing the reader to the discussion. Therefore, we would like to keep this part of the discussion.

Reviewer 4 Report

The authors' idea for the study is quite exciting and cognitive.
The study was conducted reliably and with due diligence. Interesting analyzes were carried out, and excellent statistical evaluations were made.

I have a few comments.

I find some wording quite unfortunate. For example: "The aim of this study was to evaluate whether these biological mediators were associated to the rate of improvement in the performance after training, assessed by means of a standardized test. "

It should be changed into:"The aim of this study was to evaluate whether these biological mediators changed and if it was associated to the rate of improvement in the performance after the training..."

Methods section:

There is no information on whether there were vegetarians among the subjects or whether anyone received vitamin B12, iron, or folic acid or supplements. This should be included in the text, or insert this data in the criticism of the method that it was not checked. Did the subjects have any chronic diseases? Were there any criteria for exclusion from the study? I didn't find it in the text.

In addition, I have a question about inflammation; maybe (it would be great (?) if some inflammatory parameters were assessed, such as hsRP, or even WBC in morphology, and checked if everyone was in a similar condition. Sometimes it is enough to have tooth inflammation, which also indirectly impacts study results.

In the Results section (besides no changes in the plasmatic values of Hb, iron...), "plasmatic" should be replaced by "plasma."

What did the authors mean by "T" in the sentence: p.5, line 202, "... of Vit D and T..."?

I don't understand this sentence: The BAP/U CARR ratio was reduced in mid-season (but not significantly), probable due
to the smaller increase of BAP (+1.1%) compared to U CARR (+12%). - This is quite an obvious suggestion. It explains nothing. Also probable should be replaced by probably

The discussion should be divided into subsections. Otherwise, it's tough to get through it. Maybe the authors should insert paragraphs to separate the study's limitations.

The conclusions are correct and are based on the results obtained.
Finally, I do not fully find the justification for such wording: "Monitoring the measures of oxidative stress could be a useful additional tool to suggest coaches some changes in the training and/or recovery program, which should be adequate to each player, and possibly the use of a supplementation with exogenous anti-oxidants.” What do the authors mean? How could these measurements suggest trainers change something in training? Can they explain it deeper?

Technical errors:

instead of "am" should be written (a.m.)

(kg/m2) should be replaced by (kg/m2)

(august), (February) should be changed to (August) (February)

H2O2 should be replaced by H2O2

-20oC change it to -20oC

Author Response

The authors' idea for the study is quite exciting and cognitive.
The study was conducted reliably and with due diligence. Interesting analyzes were carried out, and excellent statistical evaluations were made.

I have a few comments.

1) I find some wording quite unfortunate. For example: "The aim of this study was to evaluate whether these biological mediators were associated to the rate of improvement in the performance after training, assessed by means of a standardized test. " It should be changed into:"The aim of this study was to evaluate whether these biological mediators changed and if it was associated to the rate of improvement in the performance after the training..."

We really appreciate a lot your suggestion, and accordingly we made changes in the text.

Methods section:

2) There is no information on whether there were vegetarians among the subjects or whether anyone received vitamin B12, iron, or folic acid or supplements. This should be included in the text, or insert this data in the criticism of the method that it was not checked. Did the subjects have any chronic diseases? Were there any criteria for exclusion from the study? I didn't find it in the text.

The first Author (MA) was the medical referent for the football-team, therefore he was informed for all the supplements or drugs prescribed to athletes. As stated in the text (lines 109-110) no supplements that could influence the performance test were prescribed, even if athletes could assume spontaneously some supplements, and could have omitted to report. In the limitations we have also considered this aspect. The data were derived from a healthy, active, and diseases free population of professional athletes; none of the soccer-players reported diseases, except for those related to sport activities, for example tendinitis, muscle injury, little bone/muscle/tendons trauma, at least in the two months before the assessment (Lines96-97). At best of our knowledge, (and of the dietician that follows the team) no athletes reported to be vegetarians.

In the official sport seasons soccer-player did not report any spontaneous supplements or drugs assumption, but an omission or a recall bias could not be excluded, and this situation could potentially influence the test performance. (lines 418-421)

3) In addition, I have a question about inflammation; maybe (it would be great (?) if some inflammatory parameters were assessed, such as hsRP, or even WBC in morphology, and checked if everyone was in a similar condition. Sometimes it is enough to have tooth inflammation, which also indirectly impacts study results.

We agree with this reviewer, but unfortunately hsCRP was not in the clinical assessment, whereas WBC was not considered in the analysis since all athletes were in good clinical condition, and not significative or clinical departure from the interval range could be found in the blood count.

4) In the Results section (besides no changes in the plasmatic values of Hb, iron...), "plasmatic" should be replaced by "plasma."

Accordingly, to your suggestion we emended the text.

5) What did the authors mean by "T" in the sentence: p.5, line 202, "... of Vit D and T..."?

Thanks to point out this problem, we change all over the text Testosterone for T, and cortisol for C.

6) I don't understand this sentence: The BAP/U CARR ratio was reduced in mid-season (but not significantly), probable due to the smaller increase of BAP (+1.1%) compared to U CARR (+12%). - This is quite an obvious suggestion. It explains nothing. Also probable should be replaced by probably.

Accordingly, to your suggestion we made change in the text

7) The discussion should be divided into subsections. Otherwise, it's tough to get through it. Maybe the authors should insert paragraphs to separate the study's limitations.

We try to separate in paragraphs the discussion but it is difficult, we hope that you could endorse how we apply to your suggestion.

The conclusions are correct and are based on the results obtained.

Finally, I do not fully find the justification for such wording: "Monitoring the measures of oxidative stress could be a useful additional tool to suggest coaches some changes in the training and/or recovery program, which should be adequate to each player, and possibly the use of a supplementation with exogenous anti-oxidants.” What do the authors mean? How could these measurements suggest trainers change something in training? Can they explain it deeper?

Accordingly, to your suggestions we change the conclusion section, softening our comments, moreover we insert a phrase in the limitation section. Therefore, the text now appears:

Limitation:

…. In the official sport seasons soccer-player did not report any spontaneous supplements or drugs assumption, but an omission or a recall bias could not be excluded, and this situation could potentially influence the test performance results. We did not take into account the different distances ran by the players on the field during the matches, the speed, the sprints, the accelerations and decelerations, which vary according to their position and role in the field. Lastly the observational retrospective study design did not enable us to disentangle the causal relationship direction between OS and athletes’ performance, that could be better clarify by larger clinical trials.

Conclusion:

Training sessions and competitions induce physiological changes in soccer players, who require specific strategies to optimize their efficiency, monitor overload and prevent the risk of injuries. The main finding of this observational study is that OS is associated to the gain in physical performance between pre-season and mid-season. Indeed, the athletes with higher levels of FR and/or a lower production of anti-oxidant substances exhibit a reduced gain of physical performance after training. In conclusion, it could be useful for coaches to monitor the measures of OS in their players, as a possible harbinger of future development of overtraining syndrome.

Technical errors:

instead of "am" should be written (a.m.)

(kg/m2) should be replaced by (kg/m2)

(august), (February) should be changed to (August) (February)

H2O2 should be replaced by H2O2

-20oC change it to -20oC

We apologize for the errors and accordingly we have made corrections.

Round 2

Reviewer 1 Report

Most of the issues were corrected. According to to ethical concerns: In my opinion, the expanation of the authors is partially acceptable, however, it would be very important (and it is the task of authors) to ask the opinion of the competent ethical committe, retrospectively, as a part of the actual revision and send it to the journal.

To point 3.: I still think, that it would be useful to mark the significant results wit a * (next to the p values, as it is commonly marked in many articles in the literature), it would not be too complicated, and most of the readers appreciate it, when they do not have to look for the p values < 0.05.

Author Response

Accordingly, to the observations raised about ethic and privacy, we submitted to our local IRB the protocol and the study design. You can find attached to the submission also the IRB-CRRM opinion.

Reviewer 2 Report

I still have ethical concerns with this study. If the authors had the intention to use participants' data for research purposes, they should have had an ethics committee approve of the research (or have an ethics committee waive the requirement). This is based on section 23 of the Declaration of Helsinki - Ethical Principles: "The research protocol must be submitted for consideration, comment, guidance and approval to the concerned research ethics committee before the study begins." 

I can think of one example from a previous study that assessed historical data from a professional ice-hockey team. Before collecting this data, the researchers ensured their data collection was reviewed and approved by their ethics committee (See Quinney HA, et al. A 26 year physiological description of a National Hockey League team. Appl Physiol Nutr Metab. 2008;33(4):753-760. doi:10.1139/H08-051)

Author Response

(The authors gave the same response as above.)

Reviewer 3 Report

I have nothing further to add. Another reviewer raised the lack of ethics registration as a serious problem. This is a fatal flaw in this study and should preclude it from publication. 

Author Response

(The authors gave the same response as above.)

Reviewer 4 Report

The authors corrected the article according to my recommendations. However, unfortunately, I have to agree with the opinion of the other reviewers, I consider the lack of consent of the bioethics committee to conduct the study as a cardinal error, and the researchers' explanations do not convince me.

Suppose the study does not require the consent of the bioethics committee. In that case, the article should contain a record that the Bioethics Committee assessed the study as such that it does not require the consent of the Bioethics Committee.

However, I believe the article should not proceed further without this.

Author Response

(The authors gave the same response as above.)
